# Learning Multiple Mappings: an Evaluation of Interference, Transfer, and Retention with Chorded Shortcut Buttons

Carl Gutwin[†], Carl Hofmeister[†], David Ledo[‡], and Alix Goguey[*]

[†]University of Saskatchewan

[‡]University of Calgary

[*]Grenoble Alpes University

## ABSTRACT

Touch interactions with current mobile devices have limited expressiveness. Augmenting devices with additional degrees of freedom can add power to the interaction, and several augmentations have been proposed and tested. However, there is still little known about the effects of learning multiple sets of augmented interactions that are mapped to different applications. To better understand whether multiple command mappings can interfere with one another, or affect transfer and retention, we developed a prototype with three pushbuttons on a smartphone case that can be used to provide augmented input to the system. The buttons can be chorded to provide seven possible shortcuts or transient mode switches. We mapped these buttons to three different sets of actions, and carried out a study to see if multiple mappings affect learning and performance, transfer, and retention. Our results show that all of the mappings were quickly learned and there was no reduction in performance with multiple mappings. Transfer to a more realistic task was successful, although with a slight reduction in accuracy. Retention after one week was initially poor, but expert performance was quickly restored. Our work provides new information about the design and use of chorded buttons for augmenting input in mobile interactions.

**Keywords**: Augmented interaction; modes; chording interfaces.

**Index Terms**: H.5.m. Information interfaces and presentation (e.g., HCI)

## 1 INTRODUCTION

Mobile touchscreen devices such as smartphones, tablets, and smartwatches are now ubiquitous. The simplicity of touch-based interaction is one of the main reasons for their popularity, but touch interfaces have low expressiveness – they are limited in terms of the number of actions that the user can produce in a single input. As a result, touch interactions often involve additional actions to choose modes or to navigate menu hierarchies.

These limitations on touch input can be addressed by adding new degrees of freedom to touch devices. For example, both Android and IOS devices have augmentations that allow the user to specify the difference between scrolling and selecting: Android uses a timeout on the initial touch (i.e., a drag starts with either a short press or a long press), and some IOS devices use pressure-sensitive

carl.gutwin@usask.ca,
carl.hofmeister@usask.ca,
david.ledo@ucalgary.ca
alix.goguey@univ-grenoble-alpes.fr

screens that use different pressure levels to specify selection and scrolling [13]. Researchers have also proposed adding a wide variety of new degrees of freedom for touch devices – including multi-touch and bimanual input [16],[32],[42], external buttons and force sensors [44], back-of-device touch [3], sensors for pen state [26] or screen tilt [39],[43], and pressure sensors [8],[9].

Studies have shown these additional degrees of freedom to be effective at increasing the expressive power of interaction with a mobile device. However, previous research has only looked at these new degrees of freedom in single contexts, and as a result we know little about how augmented input will work when it is used in multiple different applications: if an augmented input is mapped to a set of actions that are specific to one application, will there be interference when the same augmentations are mapped to a different set of actions in another application?

To find out how multiple mappings for a new degree of freedom affect learning and usage for one type of augmentation, we carried out a study with a device that provides three buttons on the side of a smartphone case. The buttons can be chorded, giving seven inputs that can be used for discrete commands or transient modes. We developed three different mappings for these chording buttons for three different contexts: shortcuts for a launcher app, colour selections for a drawing app; and modes for a text-editing app. Our study looked at three issues: first, whether learning multiple mappings with the chorded buttons would interfere with learning or accuracy; second, whether people could transfer their learning from training to usage tasks that set the button commands into more complex and realistic activities; and third, whether memory of the multiple mappings would be retained over one week, without any intervening practice.

Our evaluation results provide new insights into the use of chorded buttons as augmented input for mobile devices:

- Learning multiple mappings did not reduce performance – people were able to learn all three mappings well, and actually learned the second and third mappings significantly faster than the first;
- Multiple mappings did not reduce accuracy – people were as accurate on a memory test with three mappings as they were when learning the individual mappings;
- Performance did transfer from training to more realistic usage tasks, although accuracy decreased slightly;
- Retention after one week was initially poor (accuracy was half that of the first session), but performance quickly returned to near-expert levels.

Our work provides two main contributions. First, we show that chorded button input is a successful way to provide a rich input vocabulary that can be used with multiple applications. Second, we provide empirical evidence that mapping augmented input to multiple contexts does not impair performance. Our results provide new evidence that augmented input can realistically increase the expressive power of interactions with mobile devices.

## 2 RELATED WORK

### 2.1 Increasing Interaction Expressiveness

HCI researchers have looked at numerous ways of increasing the richness of interactions with computer systems and have proposed a variety of methods including new theories of interaction, new input devices and new combinations of existing devices, and new ways of organizing interaction. Several researchers have proposed new frameworks and theories of interaction that provide explanatory and generative power for augmented interactions. For example, several conceptual frameworks of input devices and capabilities exist (e.g., [7],[19],[22],[23]), and researchers have proposed new paradigms of interaction (e.g., for eyes-free ubiquitous computing [29] or for post-WIMP devices [4],[5],[24]) that can incorporate different types of augmentation. Cechanowicz and colleagues also created a framework specifically about augmented interactions [9]; they suggest several ways of adding to an interaction, such as adding states to a discrete degree of freedom, adding an entirely new degree of freedom, or "upgrading" a discrete degree of freedom to use continuous input.

### 2.2 Chorded Text Input

Chorded input for text entry has existed for many years (e.g., stenographic machines for court reporters, or Engelbart and English's one-hand keyboard in the NLS system [6]). Researchers have studied several issues in chorded text input, including performance, learning, and device design.

A longitudinal study of training performance with the Twiddler one-handed keyboard [30] showed that users can learn chorded devices and can gain a high level of expertise. The study had 10 participants train for 20 sessions of 20 minutes each; results showed that by session eight, chording was faster than the multi-tap technique, and that by session 20, the mean typing speed was 26 words per minute. Five participants who continued the study to 25 hours of training had a mean typing speed of 47 words per minute [30]. Because this high level of performance requires substantial training time, researchers have also looked at ways of reducing training time for novices. For example, studies have investigated the effects of using different types of phrase sets in training [31], and the effects of feedback [39],[46].

Several chording designs have been demonstrated for text entry on keypad-style mobile phones. The ChordTap system added external buttons to the phone case [44]; to type a letter, the dominant hand selects a number key on the phone (which represents up to four letters) and the non-dominant hand presses the chording keys to select a letter within the group. A study showed that the system was quickly learned, and outperformed multi-tap. A similar system used three of the keypad buttons to select the letter within the group, allowing chorded input without external buttons [33]. The TiltText prototype used the four directions of the phone's tilt sensor to choose a letter within the group [43].

### 2.3 Chorded Input for Touch

Other types of chording have also been seen in multi-touch devices, where combinations of fingers are used to indicate different states. Several researchers have looked at multi-touch input for menu selection. For example, finger-count menus use the number of fingers in two different areas of the touch surface to indicate a category (with the left hand) and an item within that menu (with the right hand) [2]. Two-handed marking menus [26] also divide the screen into left and right sections, with a stroke on the left side selecting the submenu and a stroke on the right selecting the item. Multitouch marking menus [27] combine these two approaches, using vision-based finger identification to increase the number of possible combinations. Each multi-finger chord indicates which menu is to be displayed, and the subsequent direction in which the touch points are moved indicates the item to be selected. HandMarks [41] is a bimanual technique that uses the left hand on the surface as a reference frame for selecting menu items with the right hand. FastTap uses chorded multitouch to switch to menu mode and simultaneously select an item from a grid menu [17].

Other kinds of chording have also been investigated with touch devices. The BiTouch system was a general-purpose technique that allowed touches from the supporting hand to be used in conjunction with touches from the dominant hand [41]. Olafsdottir and Appert [32] developed a taxonomy of multi-touch gestures (including chords), and Ghomi and colleagues [12] developed a training technique for learning multi-touch chords. Finally, multi-finger input on a phone case was also shown by Wilson and Brewster [44], who developed a prototype with pressure sensors under each finger holding the phone; input could involve single fingers or combinations of fingers (with pressure level as an added DoF).

### 2.4 Augmenting Touch with Other Degrees of Freedom

Researchers have also developed touch devices and techniques that involve other types of additional input, including methods for combining pen input with touch [20], using vocal input [18], using the back of the device as well as the front [3], using tilt state with a directional swipe on the touch surface to create an input vocabulary [39], or using a phone's accelerometers to enhance touch and create both enhanced motion gestures (e.g., one-handed zooming by combining touch and tilt), and more expressive touch [21].

### 2.5 Augmented Input for Mode Selection

Enhanced input can also address issues with interface modes, which are often considered to be a cause of errors [35]. Modes can be persistent or "spring loaded" (also called *quasimodes* [35]); these are active only when the user maintains a physical action (e.g., holding down a key), and this kinesthetic feedback can help people remember that they are in a different mode [37].

When interfaces involve persistent modes, several means for switching have been proposed. For example, Li and colleagues [26] compared several mode-switch mechanisms for changing from inking to gesturing with a stylus: a pen button, a separate button in the non-dominant hand, a timeout, pen pressure, and the eraser end of the pen. They found that a button held in the other hand was fastest and most preferred, and that the timeout was slow and error prone [26]. Other researchers have explored implicit modes that do not require an explicit switch: for example, Chu and colleagues created pressure-sensitive "haptic conviction widgets" that allow either normal or forceful interaction to indicate different levels of confidence [9]. Similarly, some IOS devices use touch pressure to differentiate between actions such as selection and scrolling [13].

Many techniques add new sensing capabilities to create the additional modes – for example, pressure sensors have also been used to enhance mouse input [8] and pen-based widgets [32]; three-state switches were added to a mouse to create pop-through buttons [46]; and height sensing was used to enable different actions in different height layers (e.g., the hover state of a pen [11], or the space above a digital table [38]). Other techniques use existing sensing that is currently unused in an interaction. For example, OrthoZoom exploits the unused horizontal dimension in a standard scrollbar to add zooming (by moving the pointer left or right) [1].

Despite the work that has been carried out in this area, there is relatively little research on issues of interference, transfer, or retention for augmented interfaces – particularly with multiple mappings. The study below provides initial baseline information for these issues – but first, we describe the design and construction of the prototype that we used as the basis for our evaluation.

## 3 CHORDING PHONE CASE PROTOTYPE

In order to test learning, interference, and retention, we developed a prototype that adds three hardware buttons to a custom-printed phone case and makes the state of those buttons available to applications. This design was chosen because it would enable mobile use and provide a large number of states.

### 3.1 Hardware

We designed and 3D-printed a case for an Android Nexus 5 phone, with a compartment mounted on the back to hold the circuit boards from three Flic buttons (Bluetooth LE buttons made by Shortcut Labs). The Flic devices can be configured to perform various predetermined actions when pressed; Shortcut Labs also provides an Android API for using the buttons with custom software.

We removed the PCBs containing the Bluetooth circuitry, and soldered new buttons to the PCBs (Figure 1). The new pushbuttons are momentary switches (i.e., they return to the "off" state when released) with 11mm-diameter push surfaces and 5mm travel. We tested several button styles and sizes, in order to find devices that were comfortable to push, that provided tactile feedback about the state of the press, and that were small enough to fit under three fingers. This design allows us to use the Flic Bluetooth events but with buttons that can be mounted closer together. The new buttons do not require any changes to our use of the API.

The prototype is held as a normal phone with the left hand, with the index, middle, and ring fingers placed on the pushbuttons (Figure 2). The pushbuttons are stiff enough that these three fingers can also grip the phone without engaging the buttons; the fifth finger of the left hand can be placed comfortably on the phone case, adding stability when performing chorded button combinations. We also tested a four-button version, but there were too many erroneous presses because of the user needing to grip the phone. Finally, we note that the button housing on our prototype was larger than would be required by a commercial device; we estimate that the hardware could easily be built into a housing that is only marginally larger than a typical phone case.

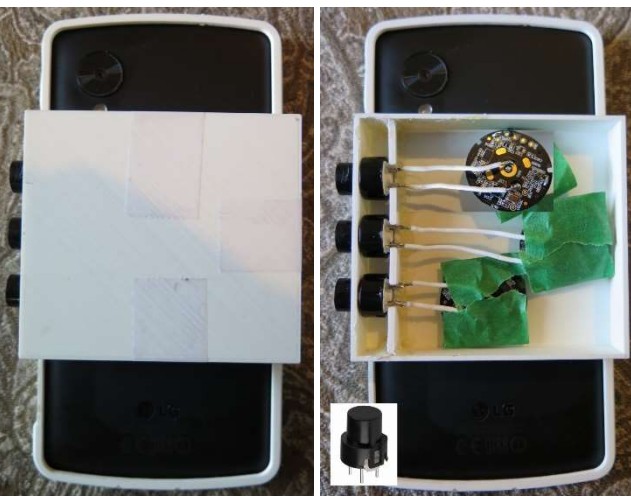

Figure 1: Chording prototype. Left: button housing. Right: Flic Bluetooth PCBs (inset shows pushbutton).

The prototype worked well in our study sessions. No participant complained of fatigue or difficulty (although we observed a few difficulties matching the timeout period, as described below). The phone case was easy to hold, and the button positions were adequate for the hand sizes of our participants. Pressing the buttons in chords did not appear to cause difficulty for any participant (although with some timing issues, as described later).

### 3.2 Software and Chord Identification

We wrote a simple wrapper library for Android to attach callback functions to the buttons through the Flic API. Android applications can poll the current combined state of the buttons through this library wrapper. Callback functions attached through the wrapper library are put on a short timer, allowing time for multiple buttons to be depressed before executing the callback. In all the applications we created, we assigned a single callback function to all the buttons; this function checks the state of all buttons and determines the appropriate behavior based on the combined state.

Identifying chords represents an interpretation problem for any input system. When only individual buttons can be pressed, software can execute actions as soon as the signal has been received from any button. When chorded input is allowed, however, this method is insufficient, because users do not press all of the buttons of a chord at exactly the same time. Therefore, we implemented a 200ms wait time (determined through informal testing) before processing input after an initial button signal – after this delay, the callback read the state of all buttons, and reported the combined pattern (i.e., a chord or a single press). Once an input is registered, all buttons must return to their "off" states before another input.

With three buttons, the user can specify eight states – but in our applications, we assume that there is a default state that corresponds to having no buttons pressed. This approach prevents the user from having to maintain pressure on the buttons during default operation.

## 4 EVALUATION

We carried out a study of our chording system to investigate our three main research questions:

- Interference: does learning additional mappings with the same buttons reduce learning or accuracy?
- Transfer: is performance maintained when users move from training to usage tasks that set the button commands into more realistic activities?
- Retention: does memory of the command mappings persist over one week (without intervening practice)?

We chose not to compare to a baseline (e.g., GUI-based commands) for two reasons: first, in many small-screen devices screen space is at a premium, and dedicating a part of the screen to interface components is often not a viable alternative; second, command structures stored in menus or ribbons (which do not take additional space) have been shown to be significantly slower than memory-based interfaces in several studies (e.g., [2][17][41]).

To test whether learning multiple mappings interferes with learning rate or accuracy, we created a training application to teach three mappings to participants: seven application shortcuts (Apps), seven colors (Colors), and seven text-editing commands (Text) (Table 1). Participants learned the mappings one at a time, as this fits the way that users typically become expert with one application through frequent use, then become expert with another.

To further test interference, after all mappings were learned we gave participants a memory test to determine whether they could remember individual commands from all of the mappings. This test corresponds to scenarios where users switch between applications and must remember different mappings at different times.

To test whether the mappings learned in the training system would transfer, we asked participants to use two of the mappings in simulated usage tasks. Colors were used in a drawing program where participants were asked to draw shapes in a particular line color, and Text commands were used in a simple editor where participants were asked to manipulate text formatting.

To test retention, we recruited a subset of participants to carry out the memory test and the usage tasks a second time, one week after the initial session. Participants were not told that they would have to remember the mappings, and did not practice during the intervening week.

Table 1. Button patterns and mappings.

| Buttons | Pattern | Color | Command | App |
|---|---|---|---|---|
| 1 | ●○○ | Red | Copy | Contacts |
| 2 | ○●○ | Green | Paste | Browser |
| 3 | ○○● | Blue | Italic | Phone |
| 1+2 | ●●○ | Yellow | Small font | Maps |
| 1+3 | ●○● | Magenta | Bold | Camera |
| 2+3 | ○●● | Cyan | Large | E-Mail |
| 1+2+3 | ●●● | Black | Select | Calendar |
| 0 | ○○○ | <panning> | <scrolling> | <none> |

### 4.1 Part 1: Learning Phase

The first part of the study had participants learn and practice the mappings over ten blocks of trials. The system displayed a target item on the screen, and asked the user to press the appropriate button combination for that item (see Figure 2). The system provided feedback about the user's selection (Figure 2, bottom of screen); when the user correctly selected the target item, the played a short tone, and the system moved on to the next item. Users could consult a dialog that displayed the entire current mapping but had to close the dialog to complete the trial. The system presented each item in the seven-item mapping twice per block (sampling without replacement), and continued for ten blocks. The same system was used for all three mappings, and recorded selection time as well as any incorrect selections (participants continued their attempts until they selected the correct item).

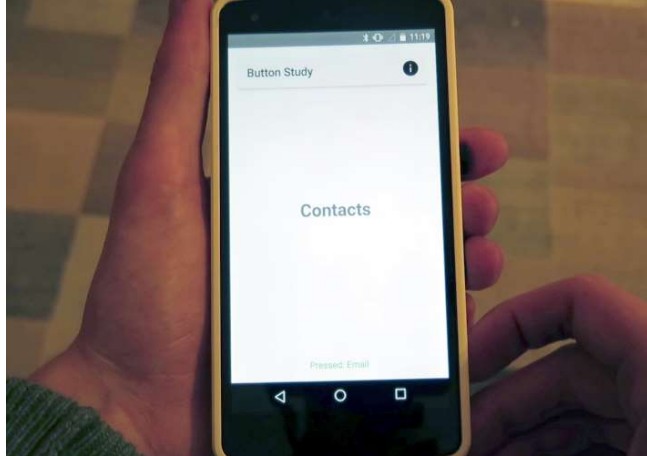

Figure 2: Training system showing Apps mapping (target at center of screen, selection feedback at bottom). Training for Color and Text mappings was similar.

### 4.2 Part 2: Usage Tasks

We created two applications (Drawing and TextEdit) to test usage of two mappings in larger and more complex activities.

*Drawing.* The Drawing application (Figure 3) is a simple paint program that uses the chord buttons to control line color (see Table 1). The application treated the button input as a set of spring-loaded modes – that is, the drawing color was set based on the current state of the buttons, and was unset when the buttons were released. For example, to draw a red square as shown in Figure 3, users held down the first button with their left hand and drew the square with their right hand; when the button was released, the system returned to its default mode (where touch was interpreted as panning). If the user released the buttons in the middle of a stroke, the line colour changed back to default grey.

For each task in the Drawing application, a message on the screen asked the participant to draw a shape in a particular color. Tasks were grouped into blocks of 14, with each color appearing twice

per block. A task was judged to be complete when the participant drew at least one line with the correct color (we did not evaluate whether the shape was correct, but participants did not know this). Participants completed three blocks in total.

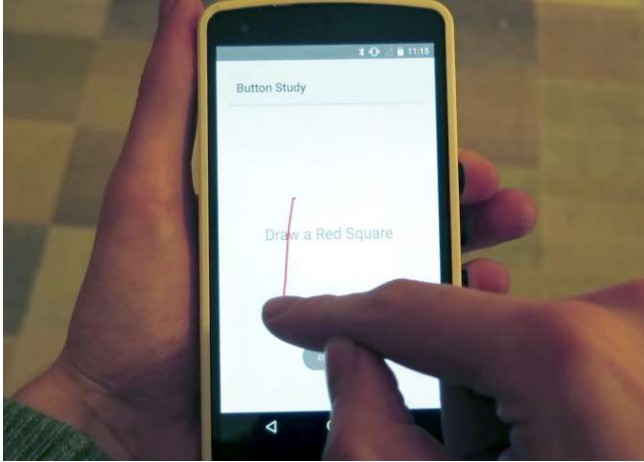

Figure 3: Drawing Task

*TextEdit.* The TextEdit application asked users to select lines of text and apply manipulations such as cutting and pasting the text, setting the style (bold or italic), and increasing or decreasing the font size. Each of these six manipulations was mapped to a button combination. The seventh action for this mapping was used for selection, implemented as a spring-loaded mode that was combined with a touch action. We mapped selection to the combination of all three buttons since selection had to be carried out frequently – and this combination was easy to remember and execute.

For each TextEdit task, the lines on the screen told the user what manipulations to make to the text (see Figure 4). Each task asked the participant to select some text and then perform a manipulation. There were six manipulations in total, and we combined copy and paste into a single task, so there were five tasks. Tasks were repeated twice per block, and there were four blocks. Tasks were judged to be correct when the correct styling was applied; if the wrong formatting was applied, the user had to press an undo button to reset the text to its original form, and perform the task again.

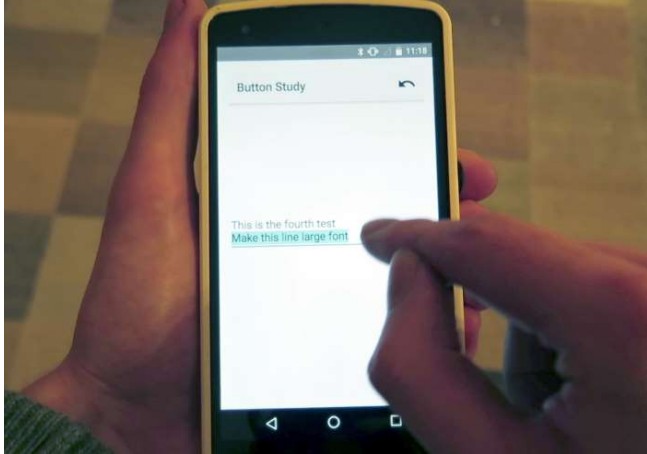

Figure 4: TextEdit task after selecting text.

### 4.3 Part 3: Memory Test

The third stage of the study was a memory test that had a similar interface to the learning system described above. The system gave prompts for each of the 21 commands in random order (Apps,

Colors, and Text were mixed together, and sampled without replacement). Participants pressed the button combination for each prompt, but no feedback was given about what was selected, or whether their selection was correct or incorrect. Participants were only allowed to answer once per prompt, and after each response the system moved to the next item.

### 4.4 Part 4: Retention

To determine participants' retention of the mappings, after the study was over we recruited 8 of the 15 participants to return to the lab after one week to carry out the memory test and the usage tasks again (two blocks of each of the drawing and text tasks). Participants were not told during the first study that they would be asked to remember the mappings beyond the study; participants for the one-week follow-up were recruited after the initial data collection was complete. The usage and memory tests operated as described above.

### 4.5 Procedure

After completing an informed consent form and a demographics questionnaire, participants were shown the system and introduced to the use of the external buttons. Participants were randomly assigned to a mapping-order condition (counterbalanced using a Latin square), and then started the training tasks for their first mapping. Participants were told that both time and accuracy would be recorded but were encouraged to use their memory of the chords even if they were not completely sure. After the Color and Text mappings, participants also completed the usage tasks as described above (there was no usage task for the Apps mapping). After completing the learning and tasks with each mapping, participants filled out an effort questionnaire based on the NASA-TLX survey. After all mappings, participants completed the memory test.

For the retention test, participants filled out a second consent form, then completed the memory test with no assistance or reminder of the mappings. They then carried out two blocks of each of the usage tasks (the Drawing and TextEdit apps had the same order as in the first study).

### 4.6 Participants and Apparatus

Fifteen participants were recruited from the local university community (8 women, 7 men, mean age 28.6). All participants were experienced with mobile devices (more than 30min/day average use). All but one of the participants was right-handed, and the one left-handed participant stated that they were used to operating mobile devices in a right-handed fashion.

The study used the chording prototype described above. Sessions were carried out with participants seated at a desk, holding the phone (and operating the chording buttons) with their left hands. The system recorded all performance data; questionnaire responses were entered on a separate PC.

### 4.7 Design

The main study used two 3x10 repeated-measures designs. The first looked at differences across mappings, and used factors *Mapping* (Apps, Colors, Text) and *Block* (1-10). The second looked at interference by analyzing differences by the position of the mapping in the overall sequence, and used factors *Position* (first, second, third) and *Block* (1-10). For the memory tests, we used a 21x3x7 design with several planned comparisons; factors were *Item* (the 21 items shown in Table 1), *Pattern* (the 7 button patterns shown in column 2 of Table 1), and *Mapping* (Apps, Colors, Text).

Dependent variables were selection time, accuracy (the proportion of trials where the correct item was chosen on the first try), and errors.

## 5 RESULTS

No outliers were removed from the data. In the following analyses, significant ANOVA results report partial eta-squared ($\eta^2$) as a measure of effect size (where .01 can be considered small, .06 medium, and > .14 large [11]). We organize the results below around the main issues under investigation: training performance when learning three different mappings, interference effects, transfer performance, and retention after one week.

### 5.1 Training: Learning Rate, Accuracy, and Effort

*Selection time.* Overall, mean selection times for the three mappings were 2626ms for Apps (s.d. 2186), 2271 for Colors (s.d. 2095, and 2405 for Text (s.d. 2193). A 3x10 (*Mapping* x *Block*) RM-ANOVA showed no effect of *Mapping* ($F_{2,28}=1.06$, p=0.36). As Figure 5 shows, selection times decreased substantially across trial blocks; ANOVA showed a significant main effect of *Block* ($F_{9,126}=42.3$, p<0.0001, $\eta^2=0.40$). There was no interaction ($F_{18,252}=0.40$, p=0.98).

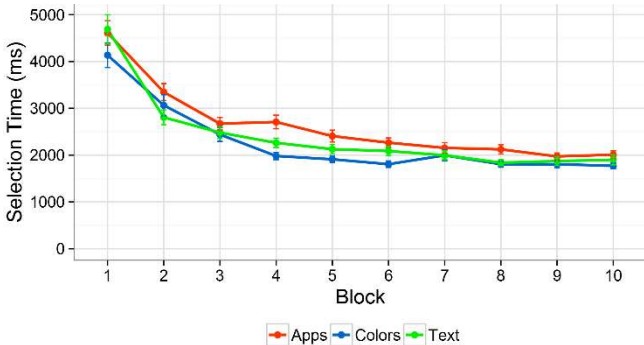

Figure 5: Mean selection time (±s.e.), by block and mapping

*Accuracy.* Across all blocks, the proportion of trials where the correct item was chosen first was 0.83 for both Apps and Colors and 0.84 for Text (all s.d. 0.37). RM-ANOVA showed no effect of *Mapping* ($F_{2,28}=0.019$, p=0.98), but again showed a significant effect of *Block* ($F_{9,126}=4.42$, p<0.001, $\eta^2=0.09$), with no interaction ($F_{18,252}=0.71$, p=0.78). Overall error rates (i.e., the total number of selections per trial, since participants continued to make selections until they got the correct answer) for all mappings were low: 0.25 errors / selection for Apps, 0.26 for Colors, and 0.24 for Text.

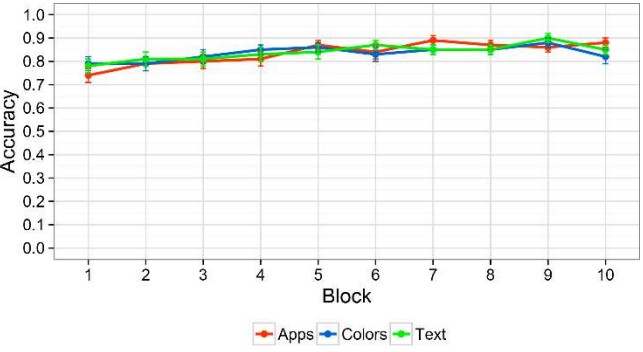

Figure 6: Mean accuracy (±s.e.) by block and mapping.

During the sessions we identified a hardware-based source of error that reduced accuracy. The 200ms timeout period in some cases caused errors when people held the buttons for the wrong period of time, when the Bluetooth buttons did not transmit a signal fast enough, or when people formed a chord in stages. This issue contributes to the accuracy rates shown above: our observations

indicate that people had the button combinations correctly memorized but had occasional problem in producing the combination with the prototype. We believe that this difficulty can be fixed by adjusting our timeout values and by using an embedded microprocessor to read button states (to avoid Bluetooth delay).

*Perceived Effort.* Responses to the TLX effort questionnaire are shown in Figure 7; overall, people felt that all of the mappings required relatively low effort. Friedman rank sum tests showed only one difference between mappings – people saw themselves as being less successful with the Apps mapping ($\chi^2=7$, p=0.030).

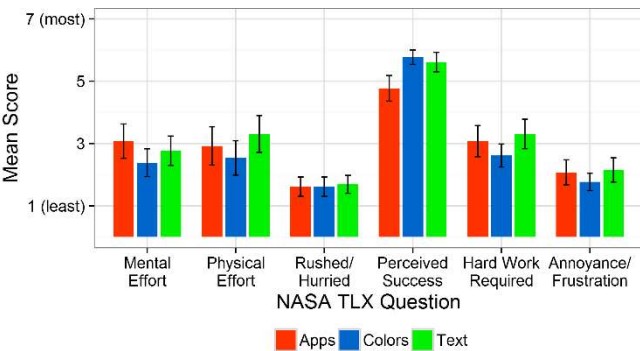

Figure 7: NASA-TLX responses (±s.e.), by mapping

In the end-of-session questionnaire, 12 participants stated that Colors were easiest to remember (e.g., one person stated "colours were easier to remember" and another said that "memorizing the colours felt the easiest").

### 5.2  Interference 1: Effects of Learning New Mappings

To determine whether learning a second and third mapping would be hindered because of the already-memorized mappings, we analysed the performance data based on whether the mapping was the first, second, or third to be learned. Figure 9 shows selection time over ten blocks for the first, second, and third mappings (the specific mapping in each position was counterbalanced).

*Selection time.* A 3x10 RM-ANOVA looked for effects of position in the sequence on selection time. We did find a significant main effect of *Position* ($F_{2,28}=19.68$, p<0.0001, $\eta^2=0.22$), but as shown in Figure 8, the second and third mappings were actually faster than the first mapping. Both subsequent mappings were faster than the first; follow-up t-tests with Bonferroni correction show that these differences are significant, p<0.01. The difference was more obvious in the early blocks (indicated by a significant interaction between Position and Block, $F_{18,252}=4.63$, p<0.0001, $\eta^2=0.14$). These findings suggest that adding new mappings for the same buttons does not impair learning or performance for subsequent mappings.

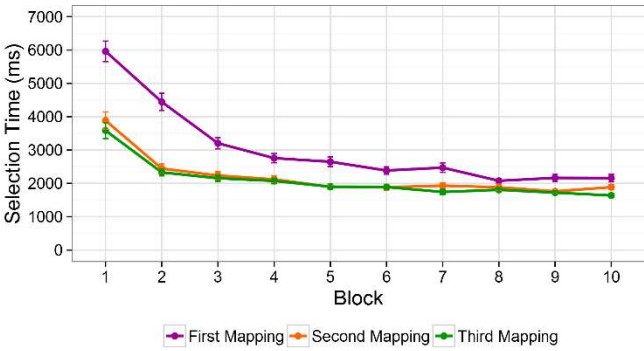

Figure 8: Mean selection time (±s.e.), by position and block.

*Accuracy.* We carried out a similar 3x10 RM-ANOVA to look for effects on accuracy (Figure 9). As with selection time, performance was worse with the first mapping (accuracy 0.8) than the second and third mappings (0.85 and 0.86). ANOVA showed a main effect of *Position* on accuracy ($F_{2,28}=7.18$, p=0.003, $\eta^2=0.072$), but with no *Position* x *Block* interaction ($F_{18,252}=1.20$, p=0.051).

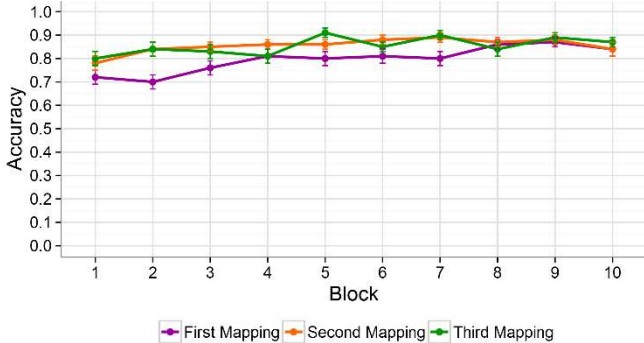

Figure 9: Mean accuracy (±s.e.), by position and block.

### 5.3  Interference 2: Memory Test with All Mappings

The third stage of the study was the memory test, in which participants selected each of the 21 commands from all three mappings, in random order. Participants answered once per item with no feedback. The overall accuracy was 0.87 (0.86 for Apps, 0.86 for Colors, and 0.89 for Text); see Figure 11. Note that this accuracy is higher that seen with the individual mappings during the training sessions.

To determine whether there were differences in accuracy for individual items, mappings, or button patterns, we carried out a 21x3x7 (*Item* x *Mapping* x *Pattern*) RM-ANOVA. We found no significant effects of any of these factors (for Item, $F_{20,240}=1.55$, p=0.067; for Mapping: $F_{2,24}=0.43$, p=0.65; for Pattern, $F_{6,12}=0.0004$, p=0.99), and no interactions.

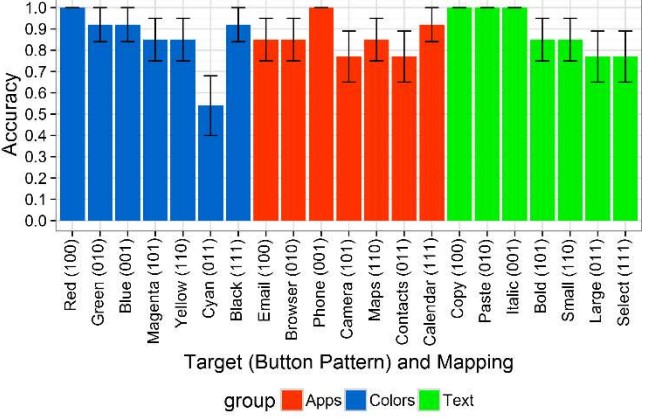

Figure 10: Memory test 1. Mean accuracy (±s.e.), by item and mapping. Button patterns shown in parentheses for each item.

Figure 10 also shows that multi-finger chords are not substantially different from single-finger button presses. Accuracy was only slightly lower with the (101) and (011) patterns than the single-finger patterns, and the one three-finger pattern (111) had an accuracy above 90% for two of the three mappings.

### 5.4 Transfer: Performance Transfer to Usage Tasks

After learning the Color and Text mappings, participants carried out usage tasks in the TextEdit and Drawing applications. Accuracy results are summarized in Figure 10 (note that the text task had four blocks, and the drawing task had three blocks). Accuracy in the usage tasks ranged from 0.7 to 0.8 across the trial blocks – slightly lower than the 0.8-0.9 accuracy seen in the training stage of the study. It is possible that the additional mental requirements of the task (e.g., determining what to do, working with text, drawing lines) disrupted people's memory of the mappings – but the overall difference was small.

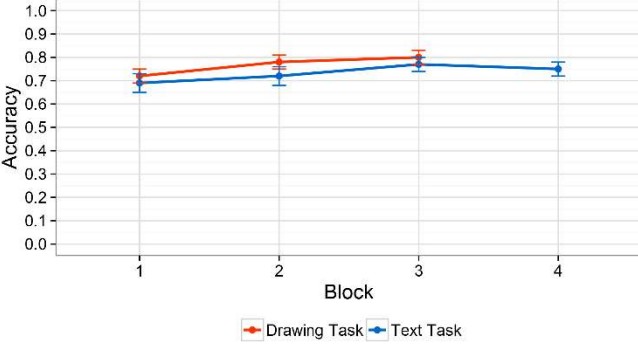

Figure 11: Mean accuracy (±s.e.), by task and block.

### 5.5 Retention: Performance After One Week

The one-week followup asked eight participants to carry out the memory test and two blocks of each of the usage tasks, to determine whether participants' memory of the mappings had persisted without any intervening practice (or even any knowledge that they would be re-tested). Overall, the follow-up showed that accuracy decayed substantially over one week – but that participants quickly returned to their previous level of expertise once they started the usage tasks. In the memory test, overall accuracy dropped to 0.49 (0.43 for Apps, 0.50 for Colors, and 0.54 for Text), with some individual items as low as 10% accuracy. Only two items maintained accuracy above 0.85 – "Red" and "Copy".

The two usage tasks (Drawing and Text editing) were carried out after the memory test, and in these tasks, participant accuracy recovered considerably. In the first task (immediately after the memory test), participants had an overall 0.60 accuracy in selection; and by the second block, performance rose to accuracy levels similar to the first study (for Drawing, 0.82; for Text, 0.70).

This follow-up study is limited – it did not compare retention when learning only one mapping, so it is impossible to determine whether the decay arose because of the number of overloaded mappings learned in the first study. However, the study shows that retention is an important issue for designers of chorded memory-based techniques. With only a short training period (less than one hour for all three mappings) appears to be insufficient to ensure retention after one week with no intervening practice; however, in an ecological context users would likely use the chords more regularly. In addition, participants' memory of the mappings was restored after only a few minutes of use.

## 6 DISCUSSION

Our study provides several main findings:

- The training phase showed that people were able to learn all three mappings quickly (performance followed a power law), and were able to achieve 90% accuracy after training;
- Overloading the buttons with three mappings did not cause any problems for participants – the second and third mappings

were learned faster than the first, and there was no difference in performance across the position of the learned mappings;
- People were able to successfully transfer their expertise from the training system to the usage tasks – although performance dropped by a small amount;
- Performance in the memory test, which mixed all three mappings together, was very strong, with many of the items remembered at near 100% accuracy;
- Retention over one week without any intervening practice was initially poor (about half the accuracy of the first memory test), but recovered quickly in the usage tasks to near the levels seen in the first sessions.

In the following paragraphs we discuss the reasons for our results, and comment on how our findings can be generalized and used in the design of richer touch-based interactions.

### 6.1 Reasons for results

People's overall success in learning to map twenty-one total items to different button combinations is not particularly surprising – evidence from other domains such as chording keyboards suggests that with practice, humans can be very successful in this type of task. It is more interesting, however, that these 21 items were grouped into three overloaded sets that used the same button combinations – and we did not see any evidence of interference between the mappings. One reason for people's success in learning with multiple button mappings may be that the contexts of the three mappings were quite different, and there were few conceptual overlaps in the semantics of the different groups of items (e.g., colors and application shortcuts are quite different in the ways that they are used). However, there are likely many opportunities in mobile device use where this type of clean separation of semantics occurs – suggesting that overloading can be used to substantially increase the expressive power of limited input.

People were also reasonably successful in using the learned commands in two usage tasks. This success shows that moving to more realistic tasks does not substantially disrupt memories built up during a training exercise – although it is likely that the added complexity of the tasks caused the reduction in accuracy compared to training. The overall difference between the training and usage environments was relatively small, however; more work is needed to examine transfer effects to real-world use.

The relatively low accuracy of our system (between 80% and 90%) is a potential problem for real-world use. The error rate in our device may have been inflated due to the timeout issue described above; further work is needed to investigate ways of reducing this cause of error. We note, however, that there are still situations in which techniques with non-perfect accuracy can still be effective (such as interface for setting non-destructive parameters and states).

Finally, the additional decay in memory of the mappings over one week may simply be an effect of the human memory system – our training period was short, and early studies on "forgetting curves" show approximately similar decay to what we observed. It is likely that in real-world settings, the frequency of mobile phone use would have provided intervening practice that would have maintained users' memory – but this issue requires further study.

### 6.2 Limitations and opportunities for future work

The main limitations of our work are in the breadth and realism of our evaluations, and in the physical design of the prototype. First, although our work takes important steps towards ecological validity for augmented input, our study was still a controlled experiment. We designed the study to focus on real-world issues of interference, transfer, and retention but the realism of our tasks was relatively low. Therefore, a critical area for further work is in testing our system with real tasks in real-world settings. The Flic software

allows us to map button inputs to actions in real Android applications, so we plan to have people use the next version of the system over a longer time period and with their own applications.

Second, it is clear that additional engineering work can be done to improve both the ergonomics and the performance of the prototype. The potential errors introduced by our 200ms timeout are a problem that can likely be solved, but the timeout caused other problems as well – once participants were expert with the commands, some of them felt that holding the combination until the application registered the command slowed them down. Adjusting the timeout and ensuring that the system does not introduce additional errors is an important area for our future work. We also plan to experiment with different invocation mechanisms (e.g., selection on button release) and with the effects of providing feedback as the chord is being produced.

An additional opportunity for future work that was identified by participants during the study is the potential use of external chorded buttons as an eyes-free input mechanism. The button interface allows people to change input modes without shifting their visual attention from the current site of work, and also allows changing tools without needing to move the finger doing the drawing (and without occluding the workspace with menus or toolbars).

## 7  CONCLUSION

Expressiveness is limited in mobile touch interfaces. Many researchers have devised new ways of augmenting these interactions, but there is still little understanding of issues of interference, transfer, and retention for augmented touch interactions, particularly those that use multiple mappings for different usage contexts. To provide information about these issues with one type of augmented system, we developed a phone case with three pushbuttons that can be chorded to provide seven input states. The external buttons can provide quick access to command shortcuts and transient modes, increasing the expressive power of interaction. We carried out a four-part study with the system, and found that people can successfully learn multiple mappings of chorded commands, and can maintain their expertise in more-complex usage tasks (although overall accuracy was low). Retention was also an important issue – accuracy dropped over one week, but was quickly restored after a short period of use. Our work provides new knowledge about the use of chorded input, and shows that adding simple input mechanisms such as chording buttons have promise as a way to augment mobile interactions.

### ACKNOWLEDGMENTS

Funding for this project was provided by the Natural Sciences and Engineering Research Council of Canada, and the Plant Phenotyping and Imaging Research Centre.

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
