# OpenReview forum: "Learning Multiple Mappings: an Evaluation of Interference, Transfer, and Retention with Chorded Shortcut Buttons"
_graphicsinterface.org/Graphics_Interface/2020/Conference — GI 2020_

### Official Review · AnonReviewer3 · 2019-12-30
**Tidy, elegant study of chord mapping overloading**

**Confidence:** 4
**Rating:** 8

**Review:**

This paper presents a study of how mobile phone users can transfer and overload gestural inputs between different contexts: in particular, the authors investigate chorded inputs for specifying colors, apps, and textual characteristics. The study described is tidy and interesting, with well-thought-out implications for design. The authors also briefly describe the physical prototype system they created for testing chorded button interactions on an Android phone.

I really enjoyed reading this paper, which asked and answered a very precise, important research question in the space of mobile interactions. While mobile interactions aren’t my specialization, I believe this work to be novel. The study design was well-suited to the task at hand (although I do have a small question about one of the mappings tested, see below). The authors’ use of plots clarified and emphasized their analysis points, making the work easy to follow and evaluate. It was really exciting to me that the authors determined that “multiple mappings did not reduce accuracy”! I enjoyed the discussion section’s discourse on how to understand and apply this result.

The technical system that supported the study was detailed at a level appropriate for replication. I am curious how the authors chose the 200ms timeout: was there a pilot study run to determine how long the timeout should be? Did they record data during the studies which would allow them to empirically determine a superior timeout? I’m also interested in how this compares to the timeout used for e.g., Twiddlers or other chorded input devices.

The mappings described by the authors were interesting, as one of them was semi-modal (colors) and two were simple inputs. The colors mapping was especially interesting to me, as it has a “natural” component to it: i.e., red + blue = magenta, just as in colour theory. I was surprised to note that this didn’t seem to affect its learnability versus the other two mappings. While I understand that the design of mappings was not a part of the goal of this study, I found myself wondering whether “natural” mappings of this kind would have different levels of memorability than “arbitrary” mappings.

Overall, I liked the paper, and I would encourage the authors to continue studying further outcomes from their work.

---

### Official Review · AnonReviewer2 · 2020-01-09
**Corded input on mobile touchscreen devices**

**Confidence:** 5
**Rating:** 5

**Review:**

The submission presents a design and evaluation of corded input for mobile devices. The proposed prototype mounts three button interface onto the side of an existing touchscreen mobile device. The user can then press on any combination of the buttons with the hand holding the device to provide input. The evaluation tests participants’ ability to provide input using the prototype to select items from predefined sets (Apps, Colors, and Commands) over 10 blocks. The results report selection time and accuracy, and workload. The submission contributes knowledge about how users can use corded input to interact with their mobile devices.

The main strength of the paper is a comprehensive investigation of the proposed design and prototype. The paper is well written and the proposed technique is interesting.

However, the paper has two main weaknesses: 1) the evaluation lacks a baseline, and 2) the accuracy of the proposed method is low.

The paper presents a thorough investigation of the proposed cording interaction technique, but it remains unclear how it compares to the existing ways users select items from a set (e.g., App icons from a menu). For example, although one would expect an expert user to be able to select applications using cording quicker than using a home menu, it is likely that they would also make more selection errors using cording. It is not clear what this tradeoff is and if the speed of selection would justify potentially lower accuracy. Thus, the paper should compare against a baseline.

The accuracy of the proposed interaction is low. Although the accuracy reaches 90% this also means that the users would make at least 1 error out of every 10 selections. This would likely frustrate the user. Thus, the paper should discuss this limitation.

In summary, the submission presents an interesting interaction technique that could potentially expand the expressiveness of interactions with mobile touchscreen devices. However, the current iteration is not ready for publication. Thus, I encourage the authors to continue this work and improve the accuracy of the interaction and show comparison with a baseline.

---

### Official Review · AnonReviewer1 · 2020-01-10
**Thorough study of interference and memorisation of chorded shortcut buttons**

**Confidence:** 4
**Rating:** 8

**Review:**

This work considers mobile devices augmented with 3 side-buttons to be used in chords, and investigates if there is interference when learning new mappings for these chords (in different applications) and how easy it is to remember them after one week. Studying additional input modalities in mobile devices (for shortcuts or to enhance existing navigation) is an important and well studied subtopic in interaction research. But as the paper indicates, there is little work looking at how overloading shortcuts could affect learning and memorisation. As such I believe it is of interest to the GI community.

I am generally positive about the paper. I particularly enjoyed how the study progresses, starting from a simple training task, moving to a complex usage task that resembles real use of mobile phones where one hand holds it (and does the chords) while the other may do detailed actions on the screen, to the two memorisation tasks (one right after the training and complex usage, one a week later). The study is well reported, with the design of experimental conditions, the procedure, counterbalancing, and results reported in detail. An additional benefit is that the software and hardware prototype (although a bit bulky) could be used in practice.

I still nevertheless believe there are some aspects in the paper that require clarification / discussion:

1. The paper shows that there is no interference in learning, on the contrary overloading mappings (learning new mappings for the same chords) seem to happen faster than learning the first mapping. Nevertheless,
inn the memorization part 1 (the one done after training and usage), I felt there was an analysis missing that would shed more light to interference. The paper tests overall memorisation after training for three mappings, but does not consider the order of presentation when reporting result. It is possible the results hide memorisation interference: for example is it possible that that commands learned or used later in the study tend to be remembered more, but this difference is hidden due to counterbalancing.

2. While the paper decided to not study a base-case condition (where no augmentation is possible), I believe this is a fair decision as augmentation is provided as a means to augment simple touch interaction. Nevertheless, what is less clear is (i) why this specific augmentation was chosen, and (ii) why it was not compared to other augmentations to see if they are similar or different in terms of interference and memorability. An additional study for such a comparison is not possible in a review round, and I believe the findings of this study are interesting enough to stand on their own. But I would like for the paper to at least expand on the specific augmentation choice (3 buttons at the side) compared to all the other possible augmentations mentioned in the related work.

3. More generally, design choices should be clarified better. At the very least it should be clarified:
- (the choice of augmentation mentioned before)
- Why 3 buttons and not two or four? How was the specific location of buttons selected? Is there prior research that suggests this design is better or is the choice based on author intuition or observation of users holding the phone? I believe the 3 fingers make sense (holding my own phone), but clearly stating if any specific methodology informed (or not) the choice would be good.
- Where does the 200msec wait time to recognize the cord come from? Is it from pilot studies, previous research? This is particularly important given the timeout issues when recognising the chords.

I appreciated the authors’ honesty in reporting the timeout issue when detecting the chords. The paper reports accuracy rates that are a bit low (around 80%) but it seems this is due to how the chord is captured (rather than true accuracy which likely is higher). The reported accuracy in the discussion though seems to be wrongly reported at 80% (instead of 80%).

As a side note, I find the motivation (abstract, intro) making a somewhat broad claim about providing results on augmentation in general. There all kinds of possible augmentations as the paper admits and saying that the findings involve “augmented input” in general (end of Abstract, Intro) is stretching the  contribution - it is not clear these findings will hold for example in bimanual interaction or pressure sensor augmentation. I suggest the authors adjust their language and use the term “chorded shortcut buttons” throughout when they talk about their contribution and findings (as they do in their title and conclusions).

Finally, I enjoyed the extensive Related Work. Nevertheless, the connection to the current work are not always clear. Eg. what have previous studies not considered or found that can be useful in the context of chorded buttons? Are there any indications that chords may be a better augmentation?

Having listed some concerns, I still believe this paper makes a good contribution to GI and should get accepted with minor clarifications (and ideally reporting of possible order effects in memorisation).

---

### Meta-Review · Area_Chair1 · 2020-01-10

**Recommendation:** Accept
**Confidence:** 4

**Metareview:**

Scores for this submission were somewhat divergent (with R2 suggesting borderline reject and R1&3 suggesting accept). All reviewers agree that the investigation is very thorough. It considers the learning of chord commands on 3-side buttons on the side of a phone: in particular it studies learning performance when overloading the mappings (ie using them for different applications), then tests them in usage tasks that are realistic, and finally considers their memorability. All reviewers found the paper well written and the topic of interest to the HCI community.

The main concerns from the most negative reviewer are the lack of a comparison to a baseline (using traditional icons + menus) and the somewhat low accuracy of the approach (80%). These are indeed valid concerns. As the paper considers for the first time the question related to learning interference and memorization of chords, there is arguably enough novelty without a baseline comparison. As for the accuracy, it is low and this should be acknowledged (although it seems that participants responses where occasionally correctly memorised, but not correctly detected, so error rates may be a bit inflated).

Given the novelty of the question asked and the good study design and reporting, I would recommend accepting the paper.

The following list of changes/clarifications would improve the paper:
- Consider whether order may affect/interfere in memorisation (R1) - if results are possible to add it would be great, else consider discussing this (in limitations/discussion).
- Explain choice of augmentation (R1). Moreover, If possible explain why it was not tested against a baseline (R2) or other augmentations (R1) -  at the very least acknowledge these as limitations.
- Explain choice of 200 msec (R1,3)
- Comment on accuracy limitations (R2) and fix accuracy reported in discussion (R1).
- Adjust a bit the language in abstract + intro (R1) when it comes to reporting findings.

---

### Decision · Program_Chairs · 2020-01-11

Accept